# Technical Note: Estimating light-use efficiency of benthic habitats using underwater O₂ eddy covariance

Karl M. Attard[1, 2] & Ronnie N. Glud[1, 3]

[1]Department of Biology, University of Southern Denmark, Odense, 5230, Denmark
[2]Tvärminne Zoological Station, University of Helsinki, Hanko, 10900, Finland
[3]Department of Ocean and Environmental Sciences, Tokyo University of Marine Science and Technology, Tokyo, Japan

*Correspondence to*: Karl M. Attard (karl.attard@biology.sdu.dk)

## 0.    Abstract

Light-use efficiency defines the ability of primary producers to convert sunlight energy to primary production and is computed as the ratio between the gross primary production and the intercepted photosynthetic active radiation. While this measure has been applied broadly within terrestrial ecology to investigate habitat resource-use efficiency, it remains underused within the aquatic realm. This report provides a conceptual framework to compute hourly and daily light-use efficiency using underwater O₂ eddy covariance, a recent technological development that produces habitat-scale rates of primary production under unaltered in situ conditions. The analysis, tested on two benthic flux datasets, documents that hourly light-use efficiency may approach the theoretical limit of 0.125 O₂ photon$^{-1}$ under low light conditions but it decreases rapidly towards the middle of the day and is typically tenfold lower on a 24 h basis. Overall, light-use efficiency provides a useful measure of habitat functioning and facilitates site comparison in time and space.

## 1.  Introduction

### 1.1 Light-use efficiency

Gross primary production can be formulated as the product of incident photosynthetic active radiation ($PAR$), the fraction of absorbed $PAR$ ($fAPAR$), and the light-use efficiency ($LUE$), that is

$$GPP = PAR * fAPAR * LUE$$ (Monteith et al., 1977). The $LUE$ indicates the efficiency with which absorbed $PAR$ is converted to $GPP$ and provides a measure of the physiological and environmental

limitation of photosynthetic production. This approach has been applied broadly within the atmospheric sciences to investigate crop yield, productivity and resource-use efficiency among terrestrial biomes using eddy covariance flux tower data (Stocker et al., 2018;Hemes et al., 2020). In aquatic environments, the *LUE* concept may be applied to both phytoplankton and benthic photosynthetic production, providing a means to compare benthic and pelagic compartments and to obtain an overall ecosystem assessment. Phytoplankton studies typically investigate the quantum yield of photosynthetic production (Falkowski, 1992), whereas benthic studies have examined *LUE* on the microscale to quantify energy budgets of photosynthetic microbial mats and symbiont-bearing corals (Al-Najjar et al., 2010;Al-Najjar et al., 2012;Brodersen et al., 2014). These microscale measurements reveal that most (> 80 %) of the incident solar energy is dissipated as heat, and conservation by photosynthesis typically is < 5 %. Despite low energy utilization, some benthic ecosystems such as coral reef symbionts seem particularly efficient at converting *PAR* to *GPP*, with *LUE* approaching the theoretical limit of 8 mol photons of *PAR* required to produce 1 mol of $O_2$ through *GPP* (or 0.125 $O_2$ photon $^{-1}$) (Brodersen et al., 2014). Studies applying the *LUE* approach to larger spatial scales of the seafloor are rare. To our knowledge there is one study using chamber incubations that employs the *LUE* approach to investigate benthic community primary production in lakes (Godwin et al., 2014), so there is much scope to port *LUE* concepts to other emerging methods.

## 1.2 Eddy covariance estimates of benthic primary production

Underwater eddy covariance (EC) is a recent technological development that has emerged as an important tool in benthic primary production studies. One of its key attributes is that it generates benthic $O_2$ fluxes at a high temporal resolution (typically ~15 min) over several days, and it does so for large

seafloor areas (10s of m$^2$, i.e. on a habitat-scale) and under unaltered *in situ* conditions (Berg et al., 2007;Berg et al., 2017). Eddy covariance thus overcomes many of the limitations of traditional methods (e.g. chamber incubations) and enables primary production rates to be measured within a wide range of benthic habitats (Chipman et al., 2016;Hume et al., 2011;Long et al., 2013;Volaric et al., 2018;Attard et al., 2019b). Additionally, the EC method can resolve very small benthic fluxes down to ~1 mmol O$_2$ m$^{-2}$ d$^{-1}$ or less (Berg et al., 2009;Donis et al., 2016), which allows reliable measurements of primary production to be made in low-activity benthic settings, such as in high-latitude environments in winter and in deep phototrophic communities (Attard et al., 2014;Attard et al., 2016). Applying the *LUE* approach to EC data will therefore provide a useful measure of the efficiency with which solar energy is converted to *GPP* on the spatial scale of whole habitats.

## 1.2 Constraining hourly and daily *GPP*

Sources of variability within EC O$_2$ fluxes can be broadly grouped into two categories, namely (1) sources that bias the measured EC flux away from the 'true' benthic flux (i.e. when EC O$_2$ flux $\neq$ benthic O$_2$ flux) due to e.g. non-steady state conditions within the benthic boundary layer and (2) 'true' temporal variability in the benthic O$_2$ exchange rate (i.e. when EC O$_2$ flux = benthic O$_2$ flux) due to e.g. flow-induced advective pore water exchange in highly permeable sediments (Table 1). Despite there being numerous sources of variability, high-quality EC fluxes often show a tight coupling to sunlight (photosynthetic active radiation, *PAR*) availability on the hourly timescale, indicating a dominant primary production signal in many aquatic systems (Berg et al., 2013;Chipman et al., 2016;Attard et al., 2014;Attard et al., 2015;Rheuban et al., 2014;Long et al., 2013;Long et al., 2015;Koopmans et al., 2020;Rovelli et al., 2017).

Under ideal conditions, the measured EC fluxes represent the balance between habitat $GPP$ and $R$. Hourly and daily $GPP$ may therefore be computed from the EC fluxes as a sum of dark and light fluxes, that is: $GPP = FLUX_{day} + \left| FLUX_{night} \right|$. It is well known that this approach provides conservative estimates of $GPP$, since $R$ typically is higher during daytime in the presence of photosynthesis (Fenchel and Glud, 2000;Hotchkiss and Hall, 2014). Indeed, several EC studies have documented lower $O_2$ effluxes in the evening than in the morning under similar light intensities (a so-called 'hysteresis'), and high $R$ rates at the onset of darkness (Rovelli et al., 2017;Rheuban et al., 2014;Koopmans et al., 2020). It is generally understood that $R$ is stimulated by $GPP$; it increases progressively throughout the day as labile photosynthates accumulate (Epping and Jørgensen, 1996;de Winder et al., 1999), and the magnitude of the hysteresis is related to the lag in the ecosystem's response (in terms of $O_2$ production through $GPP$) to changing light levels (Adams et al., 2016). While it is highly relevant to quantify daytime $R$, direct measurements are usually not available. A key requirement for computing the $LUE$ is to have reliable estimates of $GPP$. In this report we will therefore aim to provide a conceptual framework for computing hourly $GPP$ from EC fluxes, and from this, compute the $LUE$. We then test this approach on measured EC flux data.

## 2. Materials and methods

### 2.1 Eddy covariance data

This study uses a four day long EC data from Attard et al. (2014) and a three day long dataset from Attard et al. (2020). Attard et al. (2014) performed seasonal measurements at subtidal (3-22 m depth) light-exposed benthic habitats in a sub-Arctic fjord in Greenland. This study uses a dataset from a protected inlet of ~3 km$^2$ located at 3 m water depth at mean low water. The seabed had silt-sand sediments and was exposed to semi-diurnal tidal currents with flow velocities typically ranging from 2-10 cm s$^{-1}$. Attard et al. (2020) conducted their seasonal study on a 5 m deep rocky mussel reef in the Baltic Sea. Two flux datasets were selected from these two studies to represent datasets with and without flux hysteresis. Instrument setup and data processing is described in detail in these papers. In

short, the EC instrumentation consisted of a single-point acoustic velocimeter (Vector, Nortek), a fast-response $O_2$ microsensor setup (McGinnis et al., 2011), and a downwelling cosine *PAR* sensor (QCP-2000, Biospherical Instruments or LI-192, Li-Cor) mounted onto the frame. The instrument was deployed from a small research vessel and was left to collect data over several days. Benthic $O_2$ fluxes were extracted for consecutive 10- or 15-min periods using the software package SOHFEA (McGinnis et al., 2014), and the fluxes were bin-averaged to 1 h for interpretation.

The location of the interrogated area of the seafloor changes with a change in flow direction. Eddy covariance measurements typically assume no horizontal flux divergence since the measurements integrate over small-scale patchiness (Rheuban and Berg, 2013). We evaluated whether this was the case by plotting hourly *GPP* against seabed *PAR* for different flow components. The effects of flow velocity on the $O_2$ fluxes for these datasets were evaluated by Attard et al. (2014) and Attard et al. (2020), who found significant positive relationships between flow velocity and flux magnitude during day and night in Greenland, and during the night but not during the day in the Baltic Sea.

## 2.2 Computing hourly *GPP*

### 2.2.1 Defining a daytime R rate

Time series of EC fluxes were split into individual 24 h sections representing periods from midnight to midnight. Each 24 h time series was aligned with corresponding seabed *PAR* data. Daytime periods were defined as periods when $PAR > 2.0$ µmol m$^{-2}$ s$^{-1}$. Each 24 h section therefore had two night-time flux periods- the first from midnight to sunrise ($J_{N1}$), and the second from sunset to midnight ($J_{N2}$). Four options for computing the daytime *R* rate were explored. The first two approaches assumed a static *R* rate during the day whereas the third and fourth approaches assumed dynamic (time-variable) daytime

$R$. In the first approach, daytime fluxes were offset by $|\overline{J_{N1}}|$ and in the second approach, daytime $R$ was defined as an average of $J_{N1}$ and $J_{N2}$ fluxes ($|\overline{J_{N1}}| + |\overline{J_{N2}}|$). These two approaches are expected to work best when $O_2$ fluxes do not show a hysteresis. However, for other datasets that do show substantial hysteresis, this approach might underestimate $R$ (and therefore $GPP$) in the second half of the day. The

third and fourth approach attempt to correct for this by assuming a dynamic hourly daytime $R$ rate that increases progressively throughout the day. The third approach assumes a linear increase in hourly daytime $R$ with time from $|\overline{J_{N1}}|$ to $|\overline{J_{N2}}|$, whereas the fourth approach assumes that R increased with cumulative PAR. This was represented as a sigmoidal increase with time from $|\overline{J_{N1}}|$ to $|\overline{J_{N2}}|$ in concert with changes in seabed $PAR$. To calculate the shape of the sigmoidal curve for this fourth approach, the

time series of $PAR$ observations ($PAR_o$) were integrated over time and the resultant data were fitted with a sigmoidal (Boltzmann) function as:

$$\int_0^{24} PAR_o\,(t) = A_2 + (A_1 - A_2)/\left(1 + \exp\left(\frac{PAR_m - x_0}{dt}\right)\right)$$

where $A_1$ and $A_2$ were the initial and final $PAR$ values, $PAR_m$ is modelled $PAR$, $x_0$ is the centre of the curve, and $dt$ is a time constant. This function gave very tight fits to the integrated $PAR$ measurements

($R^2 > 0.99$). The fitting parameters $x_0$ and $dt$ were then used to define the sigmoidal increase in daytime respiration from $A_1$ to $A_2$ ($|\overline{J_{N1}}|$ to $|\overline{J_{N2}}|$) (Fig. 1). Hourly daytime $R$ rates were computed using this approach, and then summed with their corresponding measured daytime flux to compute the $GPP$.

*2.2.2 Light-saturation curves*

The ability of the four approaches to produce reliable estimates of hourly *GPP* was evaluated using light-saturation curves. Several mathematical formulations are available to investigate photosynthetic performance (Jassby and Platt, 1976), but benthic studies typically use linear regression or the tangential hyperbolic function by Platt et al. (1980):

$$GPP = P_m * \tanh\left(\frac{\alpha I}{P_m}\right)$$

where $P_m$ is the maximum rate of benthic gross primary production (in mmol $O_2$ m$^{-2}$ h$^{-1}$), $I$ is the near-bed irradiance (PAR; in µmol photons m$^{-2}$ s$^{-1}$), and $\alpha$ is the quasi-linear initial slope of the curve (mmol $O_2$ m$^{-2}$ h$^{-1}$ [µmol PAR m$^{-2}$ s$^{-1}$]). From these curves it is possible to derive the photoadaptation parameter $I_k$ (µmol PAR m$^{-2}$ s$^{-1}$) as $I_k = P_m/\alpha$. If we assume that hourly benthic *GPP* is predominantly driven by *PAR*, then high-quality light saturation curves for *GPP* should (a) show a high correlation with *PAR* (high $R^2$ value), and (b) have a low standard error for the fitting parameters $P_m$, $\alpha$, and $I_k$. High-quality hourly *GPP* values should also be non-negative. Non-linear curve fitting was performed in OriginPro 2020 using a Levenberg-Marquardt iteration algorithm, and the standard error of the fitting parameters was scaled with the square root of reduced chi-squared statistic.

## 2.3 Estimating light-use efficiency

### 2.3.1 Constraining the fraction of absorbed PAR (fAPAR)

Direct measurements of *fAPAR* can be made using two *PAR* sensors to resolve both incident and reflected *PAR*. In benthic environments, *PAR* absorbance typically is above 80 % of incident near-bed irradiance in sedimentary habitats and approaches 100 % in habitats with greater structural complexity (higher light scattering) such as in seagrass beds (Al-Najjar et al., 2012;Zimmerman, 2003). Therefore,

while it is advisable (and feasible) to quantify both incident and reflected *PAR* throughout the EC deployment for *LUE* estimates, the assumption that *fAPAR* = 1.0 is expected to only induce a slight bias (underestimate) to the *LUE*. Since *fAPAR* was not measured in the studies by Attard et al. (2014) and Attard et al (2020), this study assumes *fAPAR* = 1.0. To test the validity of this assumption, direct measurements of *fAPAR* were made on a separate occasion at a site with bare sediments in Oslofjord in Norway in July 2019. Here, two cross-calibrated high-quality cosine *PAR* sensors (a Biospherical QCP-2000 and a Li-cor LI-192) were affixed to a frame and placed on the seafloor at a water depth of 8 m, with the sensors located 0.5 m above the seabed. The sensors logged incident and reflected *PAR* ($\mu$mol photons $m^{-2}$ $s^{-1}$) every minute over 3 days.

*2.3.2 Computing hourly and daily light-use efficiency (LUE)*

Once the best method for computing *GPP* was identified, hourly *GPP* was converted from units of mmol $O_2$ $m^{-2}$ $h^{-1}$ to $\mu$mol $O_2$ $m^{-2}$ $s^{-1}$ and the hourly *LUE* was computed as $LUE_{hourly} = GPP_{hourly}/(PAR_{hourly} * fAPAR)$, with units of $O_2$ $photon^{-1}$. Similarly, daily *GPP* (mmol $O_2$ $m^{-2}$ $d^{-1}$), computed as $GPP = FLUX_{day} + |FLUX_{night}|$, and daily integrated *PAR* (mmol photon $m^{-2}$ $d^{-1}$) were used to compute daily *LUE* ($O_2$ $photon^{-1}$) as $LUE_{daily} = GPP_{daily}/(PAR_{daily} * fAPAR)$.

## 3. Results and discussion

### 3.1 Effects of flow direction

The embayment in Greenland had a semidiurnal tidal signal i.e. two high and two low tides every day, with the two predominant flow directions (100-150° and 190-230°) accounting for 90 % of the fluxes.

In the Baltic Sea, the flow direction was more variable with 5 flow directions each accounting for 15-30 % of the fluxes. Despite the fluxes originating from different parts of the seafloor, the flow direction did not have a substantial impact on hourly *GPP*, indicating that the eddy covariance measurements adequately integrated over habitat patchiness (Fig. 2).

## 3.2 Hourly *GPP* and light-saturation curves

In the four-day dataset from Greenland (Attard et al., 2014), hourly *GPP* ranged from 0 to 8 mmol $O_2$ $m^{-2}$ $h^{-1}$ under maximum daytime irradiance of up to 500 μmol photons $m^{-2}$ $s^{-1}$. Hourly *GPP* measured in the first half of the day were very similar to rates resolved in the second half of the day under similar *PAR* intensities, indicating no substantial flux hysteresis (Fig. 3). Hourly *GPP* showed a tight correlation with seabed *PAR*, with $R^2$ values for the light-saturation curves ranging from 0.83 to 0.93 (Fig. 3). Overall, the highest $R^2$ values for the light-saturation curves for this dataset were achieved using a static daytime *R* rate which was defined as an average of all night-time fluxes ($\overline{|J_{N1}| + |J_{N2}|}$).

This approach achieved $R^2$ values in the light-saturation curves that were up to 10 % higher than when *R* was defined using the first night-time period alone ($\overline{|J_{N1}|}$). Light saturation began to occur at 20-30 % of peak daily irradiance, and no photoinhibition at high irradiance was observed. The lowest light saturation ($I_k$) and the highest alpha ($\alpha$) were measured during the day with the lowest light intensities (day two), suggesting potential low light acclimation (Fig. 3).

In the EC dataset from the Baltic Sea, a clear hysteresis was observed in the $O_2$ fluxes. Hourly $O_2$ fluxes in the second half of the day were up to 4-fold lower than within the first half of the day under similar irradiance levels. Light-saturation curve $R^2$ values varied depending on the method used to define the

daytime $R$ rate (Fig. 4). In all three days from this dataset, the highest $R^2$ values were obtained using dynamic daytime $R$ rates defined as either a linear or sigmoidal increase with time. These two approaches produced *GPP* estimates with the best quality: all hourly *GPP* values were positive, and the fitting parameters $P_m$, $I_k$ and $\alpha$ had the lowest standard errors (Fig. 4). While $P_m$ and $\alpha$ showed good agreement between the four methods, static $R$ approaches tended to overestimate the $I_k$ and underestimate $\alpha$ since hysteretic fluxes tend to bias light-saturation curves towards linearity. Following the correction, the light-saturation parameter $I_k$ decreased and the $\alpha$ increased by ~20%. This indicates that the curve becomes less linear-like, which is what we would expect when we correctly account for the minor hysteresis that we encountered. We note that other studies have documented a much larger hysteresis than what we observe at the mussel bed (Rheuban et al., 2014;Rovelli et al., 2017).

Hourly *GPP* computed using sigmoidal increases in daytime $R$ for the Baltic Sea dataset ranged from 0 to 7 mmol $O_2$ m$^{-2}$ h$^{-1}$ under *PAR* levels of up to 350 $\mu$mol photons m$^{-2}$ s$^{-1}$ (Fig. 3). Light-saturation curves provided high $R^2$ values for day 1 and day 3 of 0.83 and 0.81. The light-saturation curve for day 2 converged to a linear fit with an $R^2$ of 0.94 (Fig. 5).

## 3.3 Light-use efficiency

Hourly *LUE* estimates for the two datasets indicated high *LUE* of up to 0.09 $O_2$ photon $^{-1}$ under light-limiting conditions of < 20 $\mu$mol *PAR* m$^{-2}$ s$^{-1}$ (Fig. 6). Light-use efficiency declined quasi-exponentially with time (and *PAR*) to around one-tenth of the value by the middle of the day, and then it increased again towards sunset to *LUE* values comparable to the morning. This observation is consistent with the microsensor and benthic chamber studies by Al-Najjar et al. (2012), Brodersen et al. (2014) and

Godwin et al. (2014) who document maximum *LUE* under light-limiting conditions and a decline in *LUE* under high irradiance levels typical of the middle of the day. Phytoplankton studies have similarly documented high *LUE* (~85 % of theoretical limit) under light-limiting conditions (Sosik, 1996).

High hourly *LUE* under low light intensities was a general feature of both datasets, but it was not universal across all times of the day. The expected patterns were largely consistent in the dataset from Greenland (Fig. 3) but less so in the dataset from the Baltic Sea (Fig. 5, Fig. 6), where we often observed lower *LUE* than we would expect under light-limiting conditions. Directional differences were rather small for both datasets (Fig. 2), so these discrepancies could instead reflect other environmental differences, such as the availability of nutrients for *GPP* at these two contrasting sites (sedimentary versus rocky).

Daily *LUE* estimated as the ratio between $GPP_{daily}$ and $PAR_{daily}$ (both in mmol $m^{-2}$ $d^{-1}$) ranged from 0.008 to 0.013 $O_2$ $photon^{-1}$ in Greenland and was 0.006 to 0.007 $O_2$ $photon^{-1}$ in the mussel bed dataset from the Baltic Sea (Fig. 7). This indicates that the soft sediment habitat in Greenland had higher photosynthetic efficiency than the rocky mussel bed in the Baltic Sea on a daily timescale for the investigated data. However, in all cases daily *LUE* is at least tenfold lower than the theoretical limit of 0.125 $O_2$ $photon^{-1}$.

The *LUE* values presented in this study are expected to be underestimated due to the assumption of *fAPAR* = 1.0 (i.e. by assuming that all incident *PAR* is absorbed by the seabed). A fraction of the incoming irradiance is reflected and thus is not available for photosynthesis. Reflected *PAR* ranged from 17.5 % to 1.9 % in the study on microbial mats by Al-Najjar et al. (2012) and was up to 12 % in the coral symbiont study by Brodersen et al. (2014). Direct measurements of *fAPAR* were not available for

the datasets used in this study, but measurements from a bare sediments site in Oslofjord indicated reflected *PAR* on the order of 8-10 % (Fig. 8). It is therefore likely that the *LUE* estimates presented in this study are underestimated by ~10 %.

## 4. Conclusion

A key requirement of the *LUE* approach is high-quality *GPP* data. Despite there being numerous potential obstacles to obtaining this data (Table 1), a growing number of eddy covariance studies document tight relationships between hourly fluxes and sunlight availability in a wide array of aquatic habitats such as in sediment deposits, seagrass canopies, coralline algal beds and coral reefs (Berg et al., 2013;Chipman et al., 2016;Attard et al., 2014;Attard et al., 2015;Rheuban et al., 2014;Long et al., 2013;Long et al., 2015;Koopmans et al., 2020;Rovelli et al., 2017). In this study, $R^2$ values for light-saturation curves ranged from 0.83 to 0.94 indicating a predominant primary production signal, and this gives credence to applying the *LUE* approach.

Constraining the daytime *R* rate on an hourly timescale is clearly a challenge, especially on the spatial scales included within eddy covariance measurements. Assuming a linear or sigmoidal increase in *R* with time is consistent with observations of accumulating leached photosynthates such as carbohydrates that stimulate daytime *R* (de Winder et al., 1999;Epping and Jørgensen, 1996); however, more experimental data are required to investigate these assumptions in detail. The theoretical limit *LUE* ratio of 0.125 $O_2$ photon$^{-1}$ provides an upper constraint on the *GPP* that is possible for given *PAR* level. Hourly *LUE* at the start and at the end of the day often approached the theoretical limit (Fig. 3), so it is unlikely that the *GPP* rates in these datasets were substantially underestimated.

Light-saturation curves are a useful tool to evaluate flux hysteresis and ways to correct for this. There are several considerations when computing hourly *GPP* that will influence both the $R^2$ value as well as the fitting parameters $P_m$, $\alpha$ and $I_k$. Since these parameters hold real-world significance (i.e. they are not just operators within the mathematical expression; Jassby and Platt (1976)) it is important to consider

5   factors that may introduce bias.

Overall, the *LUE* approach provides a useful means to compare photosynthetic performance of submerged habitats on hourly and daily timescales. This provides opportunities to generate hypotheses about the importance of habitat structure (e.g. organization of photosynthetic elements) and other factors that influence benthic *GPP* such as epiphytes, grazing, nutrient availability, temperature and

10   current strength (Elser et al., 2007;Mass et al., 2010;Brodersen et al., 2015;Tait and Schiel, 2011). In terrestrial environments, this approach has been used to investigate the effects of biodiversity and biodiversity loss on habitat productivity. Similar analyses ported to the aquatic realm would constitute timely studies.

## 5. Data availability

All presented data are openly available from the Dryad Digital Repository at https://doi.org/10.5061/dryad.xwdbrv1bv

## 6. Author contribution

KMA & RNG conceived the idea, KMA collected and processed the data. KMA wrote the manuscript with input from RNG.

## 7. Acknowledgements

We are grateful to our colleagues at the Greenland Climate Research Centre in Nuuk, Greenland, at the Tvärminne Zoological Station in Finland, and at the Norwegian Institute for Water Research in Norway (NIVA; Dr Kasper Hancke) for their help with fieldwork. This work was supported by the Walter and Andreé de Nottbeck Foundation, the Academy of Finland (grant agreement numbers 283417 and 294853), Denmark's Independent Research Fund (FNU 7014-00078), and the Research Council of Norway (HAVBRUK2, KELPPRO).

## 8. Competing interests

The authors declare no competing interests.

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

Table 1: Sources of EC flux variability can be broadly grouped into two categories: (1) sources that bias the measured EC flux away from the 'true' benthic flux (i.e. when EC $O_2$ flux ≠ benthic $O_2$ flux) and (2) 'true' temporal variability in the benthic $O_2$ exchange rate (i.e. when EC $O_2$ flux = benthic $O_2$ flux)

| EC $O_2$ flux ≠ benthic $O_2$ flux | Reference | EC $O_2$ flux = benthic $O_2$ flux | Reference |
|---|---|---|---|
| Non steady-state conditions within the benthic boundary layer | (Holtappels et al., 2013;Brand et al., 2008) | Changes in diffusive boundary layer thickness in cohesive sediments | (Kuhl et al., 1996) |
| Sensor stirring sensitivity | (Holtappels et al., 2015) | Pore-water advection in permeable sediments | (Cook et al., 2007;McGinnis et al., 2014) |
| Surface wave influence | (Berg et al., 2015;Reimers et al., 2016) | Diel fauna activity | (Wenzhofer and Glud, 2004) |
| Sensor response time | (McGinnis et al., 2008;Berg et al., 2015) | Sediment resuspension | (Toussaint et al., 2014), Camillini et al. In review |
| Internal plant $O_2$ storage, canopy storage, or bubbling | (Attard et al., 2019a;Rheuban et al., 2014;Long et al., 2020) | Oxidation of anaerobic metabolites in sediments | (Fenchel and Glud, 2000) |
| | | Nutrient availability | (Elser et al., 2007) |
| | | Photosynthesis-coupled respiration | (Epping and Jørgensen, 1996) |
| | | Acclimation of the photosynthetic system | (Ralph et al., 2002) |

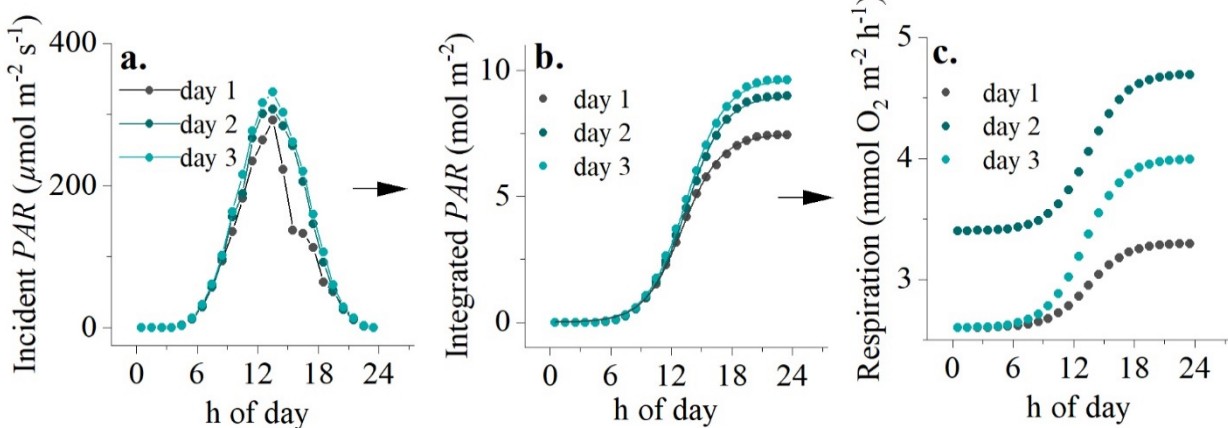

Fig. 1: The process used to derive daytime respiration rates for datasets impacted by hysteresis. (a) Incident seabed *PAR*, (b) integrated *PAR*, and (c) estimated hourly respiration rates. The line of best fit to the data in (b) is a Boltzmann function ($R^2 >$ 0.99). The fitting parameters were used to determine the shape of the respiration curve in (c) from night-time flux periods $|J_{N1}|$ to $|J_{N2}|$ (see text).

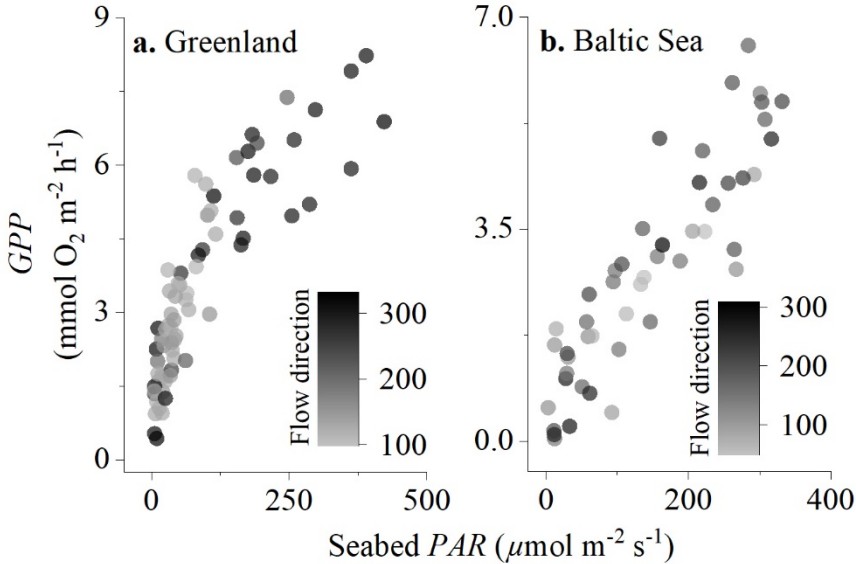

Fig. 2: Comparison of eddy covariance fluxes for (a) the sedimentary embayment in Greenland and (b) the rocky mussel reef in the Baltic Sea. Flow direction (instrument degrees) illustrates that different parts of the seafloor were included in the measurements, but this did not substantially impact the magnitude of the fluxes.

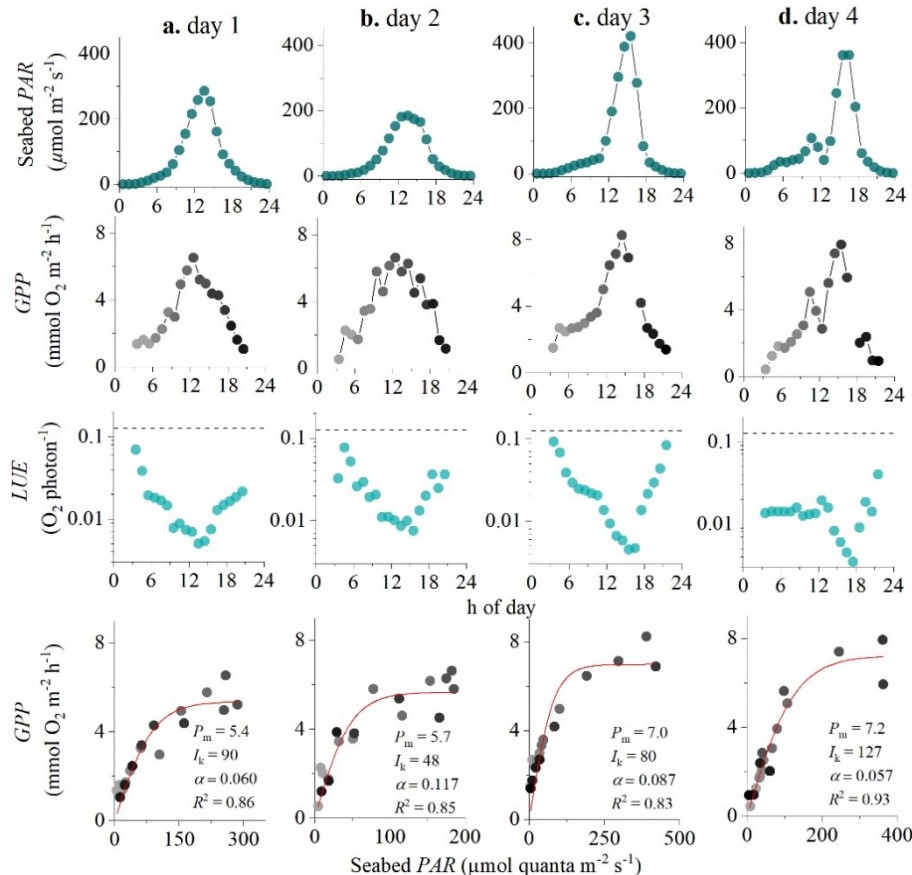

Fig. 3: Eddy covariance data measured over four consecutive days in the sedimentary embayment in Greenland showing seabed PAR (top panels), hourly *GPP* (second row), estimated hourly *LUE* (third row; dashed lines indicate theoretical limit), and corresponding light-saturation curves (bottom panels). Symbols in the second and fourth rows are colour-mapped by h of day. Light-saturation curves are fitted to the data showing the maximum rate of *GPP* ($P_m$, mmol $O_2$ m$^{-2}$ h$^{-1}$), the photoadaptation parameter $I_k$ (µmol PAR m$^{-2}$ s$^{-1}$), the initial slope of the curve $\alpha$, and the coefficient of determination ($R^2$).

Data modified from Attard et al. (2014).

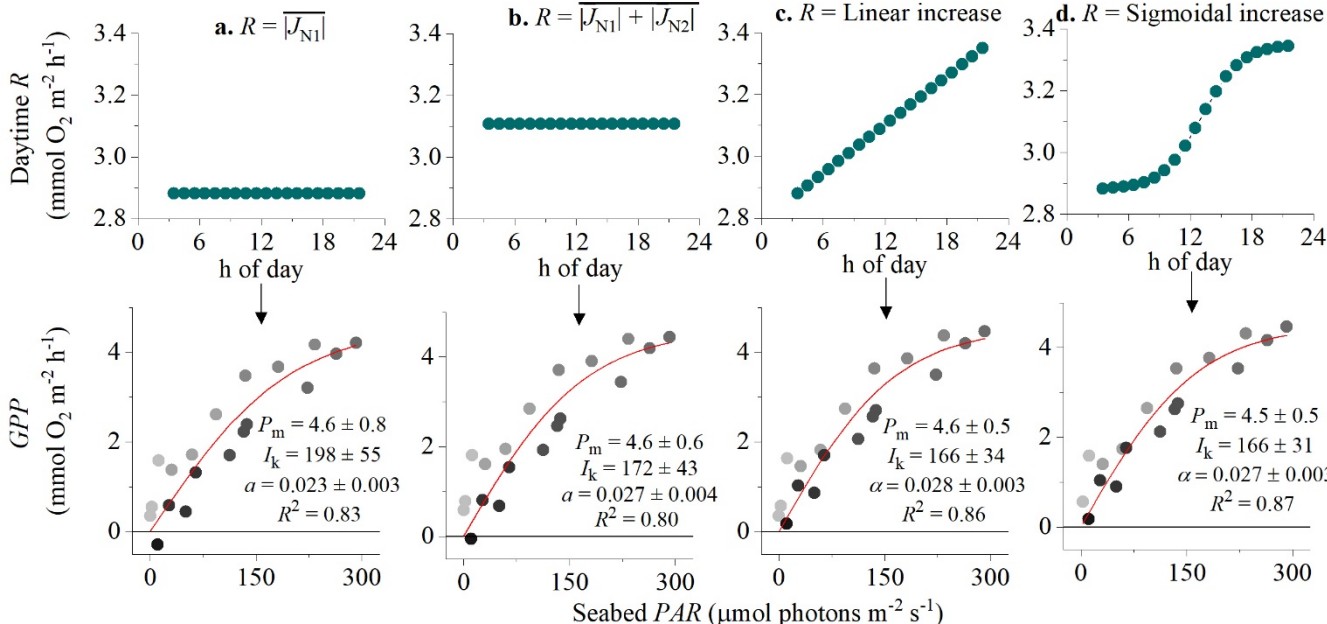

Fig. 4: Different approaches for defining the $R$ rate during the day (and therefore the hourly $GPP$) from eddy covariance fluxes showing hysteresis: (a) $R$ = average flux for the first night-time flux period ($\overline{|J_{N1}|}$), (b) $R$ = average flux for both night-time periods ($\overline{|J_{N1}| + |J_{N2}|}$), (c) $R$ increases linearly from $\overline{|J_{N1}|}$ to $\overline{|J_{N2}|}$, and (d) $R$ increases from $\overline{|J_{N1}|}$ to $\overline{|J_{N2}|}$ following a sigmoidal curve. Bottom panels show corresponding light-saturation curves and fitting parameters for the maximum rate of $GPP$ ($P_m$, mmol $O_2$ m$^{-2}$ h$^{-1}$), the photoadaptation parameter $I_k$ (µmol PAR m$^{-2}$ s$^{-1}$), the initial slope of the curve $\alpha$, and the coefficient of determination ($R^2$). Symbols in bottom panels are colour-mapped by h of day. Data modified from Attard et al. (2020).

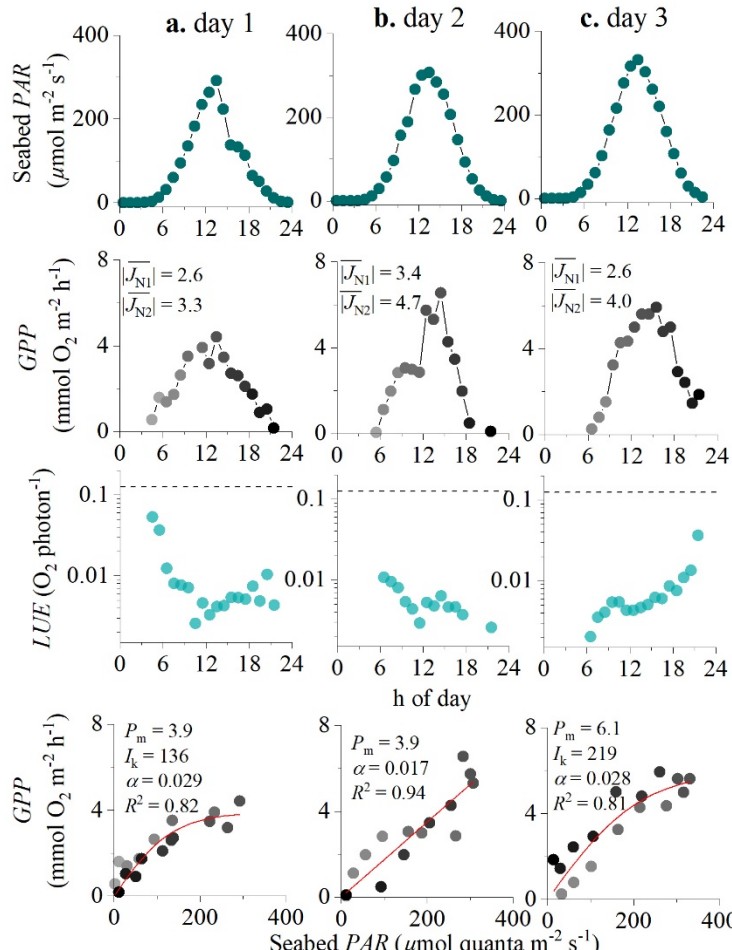

Fig. 5: Eddy covariance data measured over three consecutive days in the rocky mussel reef in the Baltic Sea showing seabed *PAR* (top panels), hourly *GPP* (second row), estimated hourly LUE (third row; dashed lines indicate theoretical limit), and corresponding light-saturation curves (bottom panels). Symbols in the middle and bottom panels are colour-mapped by h of day. Light-saturation curves are fitted to the data showing the maximum rate of *GPP* ($P_m$, mmol $O_2$ m$^{-2}$ h$^{-1}$), the photoadaptation parameter $I_k$ (µmol PAR m$^{-2}$ s$^{-1}$), the initial slope of the curve $\alpha$, and the coefficient of determination ($R^2$). For each 24 h period, average fluxes for the first and second night-time periods are shown in the middle panels ($|\overline{J_{N1}}|$ and $|\overline{J_{N2}}|$, in mmol $O_2$ m$^{-2}$ h$^{-1}$). Data modified from Attard et al. (2020).

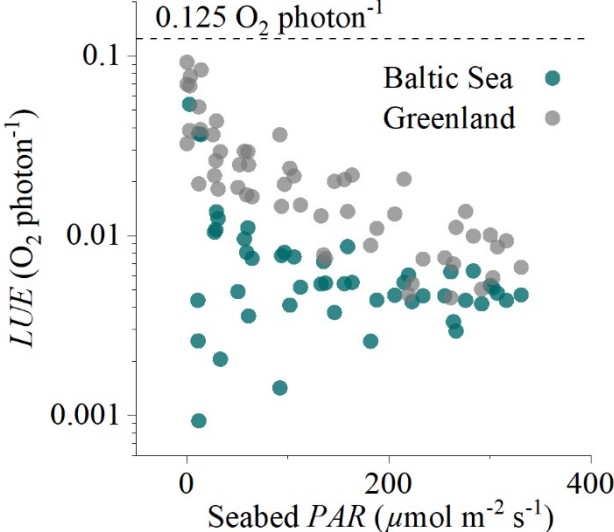

Fig. 6: Hourly light-use efficiency (*LUE*, log-axis) plotted against incoming irradiance (seabed *PAR*) for the two eddy flux datasets collected in the sedimentary embayment in Greenland and the rocky mussel reef in the Baltic Sea. The broken line indicates the theoretical limit of 0.125 $O_2$ photon $^{-1}$.

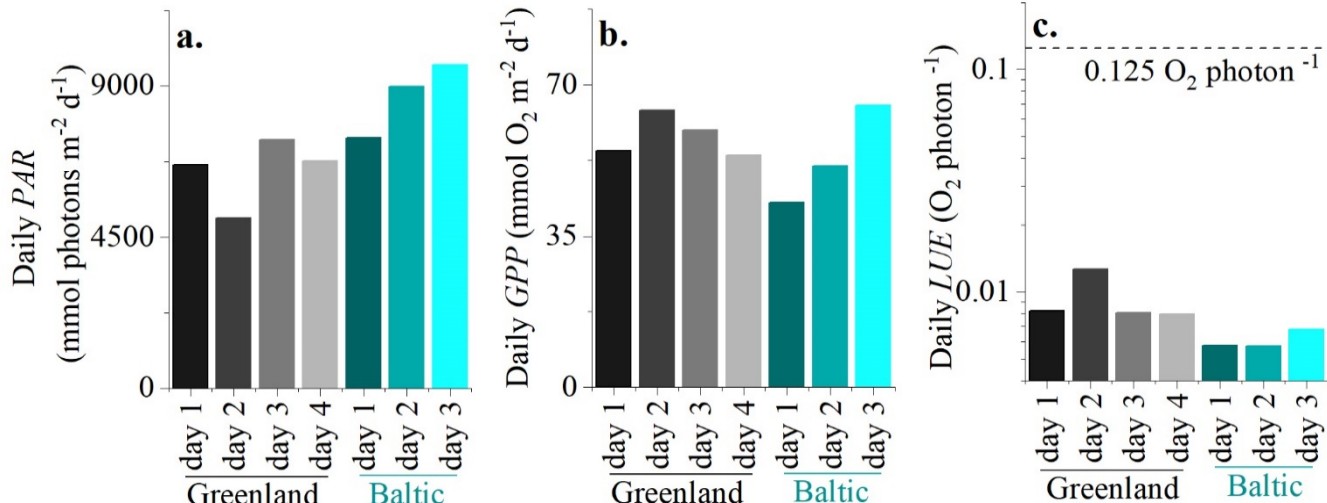

Fig. 7: (a) Daily seabed *PAR*, (b) daily benthic *GPP*, and (c) daily *LUE*. The broken line in (c) indicates the theoretical limit of 0.125 $O_2$ photon $^{-1}$.

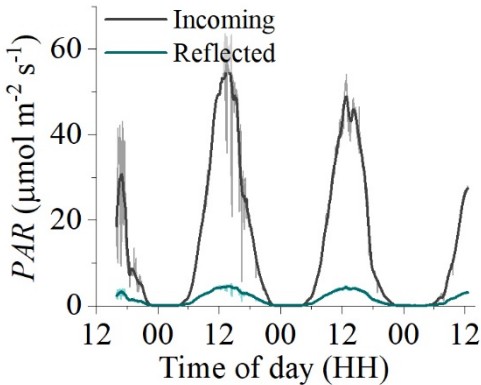

Fig. 8: Measurements of incident and reflected seabed *PAR* made using two cosine *PAR* sensors over a habitat with bare sediments at 8 m depth in Oslofjord in July 2019. Reflected *PAR* was typically 8-10 % of incident *PAR*, indicating that ~90 % of incident *PAR* was absorbed by the benthos.

