# Peer review of "Technical Note: Estimating light-use efficiency of benthic habitats using underwater O2 eddy covariance"

_Biogeosciences, 2020_

## Referee Comment (RC1) · Anonymous Referee #1 · 2 May 2020

This is a well-written, interesting technical note but the data analysis needs further explanations.

In this study, the authors use the underwater eddy covariance technique to measure oxygen flux in shallow coastal environments where light reaches the seafloor. From these fluxes, they compute hourly and daily light-use efficiency of the phototrophic benthic community. One of the key findings is that the hourly light-use efficiency may approach the maximum theoretical limit and that it decreases rapidly towards the middle of the day. These are nice results that are also supported by previous work by Berg and colleagues and should be of interest for the readers of biogeosciences. Light

use efficiency is a useful parameter for characterizing and comparing shallow benthic habitats and for assessing environmental change. In a time when coastal water quality is deteriorating globally, a technique allow evaluation of the activity of the phototrophic benthic community is very helpful.

I propose expanding the discussion of the calculations of GPP and R and their limitations. Gross primary production (GPP, here total oxygen produced through photosynthesis) was calculated as the sum of the daytime measured net oxygen production and the oxygen consumed through respiration (R) at night. As pointed out by the authors, daytime respiration typically exceeds nighttime respiration, but daytime respiration could not be measured directly in this study. Thus, four different daytime respiration rates were calculated, two static rates and two dynamic rates (linear or sigmoid increases) to determine the respiration behavior that would fit best with the measured data. The accuracy of the determination of R and GPP defines the quality of the light use efficiency estimates that are at the center of this study. In a tidal regime, the eddy covariance instrument may not interrogate the same area of the seafloor during day and night, and thereby produce nighttime R data that are not representative, even after some corrections, of the area producing the daytime flux data. The actual differences in R may be small, however, R then represents a best guess, not a known flux.

Another point that could be addressed in more detail are the other controlling factors of benthic photosynthesis besides light intensity, e.g. the spectral composition of the light, roles of grazers, nutrient availability, temperature and current strength.

In figure 1, the data could be interpreted differently, i.e. further increase of the light-saturation curves with increasing light. These are four consecutive days of measurements, and the curves of the third and fourth days increase until 300 PAR at least if not farther.

In figure 2, second panel, N1+N2 should be changed to be (N1+N2) average.

Although R is about 20% higher in plot 2 of figure 2, GPP is almost identical, and an

explanation for this unexpected behavior would be helpful. Similarly, as R increases over time in fig. 2, c and c, one could expect that the curvature of the light-saturation curves would increase but it does the opposite. All four GPP plots in fig. 2 are nearly identical suggesting that magnitude and dynamics of R have little influence on GPP. This is counterintuitive as in many coastal environments R reaches similar magnitude as GPP (as also seen in figure 2) and also follows dynamics that may be similar to those of the GPP (as in fig. 2 d). This needs more detailed explanation.

---

## Referee Comment (RC2) · Anonymous Referee #2 · 11 May 2020

**1. General comments**

This Technical Note presents a technique that is novel to aquatic ecosystem research and may be highly useful. The approach is simple, but appears to be adequate for this first application. The manuscript is well written and concise, but minor changes can improve it. Specifically, the introduction could be improved by focusing, from the first sentence, on the scientific value that light use efficiency measurements can provide. This can be accomplished largely by re-arranging text, but it will also require more citations. In addition, the Boltzmann equation can be more clearly presented. Finally, two additional figures would also improve the manuscript. The first would be of the

sigmoidal fit to PAR data. The second would be of a diurnal time series of light use efficiency. A brief discussion of the data quality, patterns, and implications of that last, proposed, figure would be a valuable addition.

2. Specific comments

2.1 Introduction

The introduction focuses on the eddy covariance technique, but LUE is important well beyond eddy covariance. The manuscript would address a larger audience if Section 1.3 can be adapted so that it is a suitable first section. The approach you suggest is exciting. It can be used to investigate the physiological and environmental limitation of photosynthetic production. Those measurements will be of interest outside the eddy covariance community. Former Sections 1.1 and 1.2, that focus on eddy covariance, can be adapted to follow the new Section 1.1. Those sections would show that the tight relationship of eddy covariance measurements to PAR suggests that they will be adequate to resolve LUE in aquatic ecosystems.

In the new first section, please also introduce quantum yield measurements. Use caution. In aquatic literature, the term commonly refers to the quantum yield of the photochemistry of photosystem II. Those measurements don't directly compare to LUE. However, there has also been extensive research on the quantum yield of phytoplankton photosynthetic production. Those measurements can be compared directly to your own. Both measurements can reveal environmental and physiological limitations of photosynthesis. Falkowski and Raven (1997) provide a useful summary in Chapter 3.

2.2 Materials and methods

There are four cumulative PAR terms in the equation (left side, A1, A2, and PAR). For clarity, I suggest writing that A1, A2, xo and dx are all fitting parameters. PAR on the left and PAR on the right can be discriminated with different subscripts. Preferably, the subscript would make it clear that one is observed and the other is predicted. I also

suggest that x0 and dx would be more clearly represented with as "t" instead of "x," because they are measures of time.

To clarify how the sigmoidal fit is used, please include a figure that shows the fit to cumulative PAR. With an additional y-axis, this figure could also show a representative increase in daytime R that is predicted with it. Accordingly, please also include evidence that N2 often exceeds N1.

2.3 Results and discussion

Please also add a figure to show a diurnal time series of changes in LUE alongside PAR and GPP. Ik is the irradiance at which the rate of photon absorption matches the maximum turnover rate of photochemistry (Falkowski and Raven, p. 200). Your results can be used to examine if Ik is a point of interest in the diminishment LUE as PAR increases. Assuming the resulting figure is noisy, please also consider techniques that could improve resolution of diurnal variation in LUE.

3. Technical corrections

Page 2 Line 20: "offsetting daytime fluxes by the dark rate." "Offsetting" is a vague term. Can you use a more specific one? I suggest "sum" or "summation." You use "offset" again on page 5, line 11.

Page 3 line 7: "magnitude of hysteresis is related to light history." "light history" is also vague term. Could you be more specific?

Page 5 Defining a daytime R rate. It is not clear that N1 and N2 are measurements of flux, as opposed to periods of time. I suggest clarifying that they are flux. By convention, J is often used for flux.

Page 5 line 17: "...whereas the fourth approach assumed a sigmoidal increase with time." I'd suggest more specifics here too. Perhaps "...whereas the fourth approach assumed that R increased with cumulative PAR. This was represented with a sigmoidal increase..."

Page 7 line 10: Instead of "assuming," I suggest "the assumption that."

Page 7 line 21 PARhourly *fAPAR should be enclosed by parentheses. Again on page 8, line 2.

Page 8, line 3 Check the cases of section headings. You may want "Results and discussion"

Page 8, line 5 Please describe the habitat types of the dataset from Greenland.

Page 9 line 11: In Figure 4 there are a handful of measurements with a very low LUE (0.001 to 0.004). Could you say a few words about these?

Page 9, line 19. Replace "∼" with approximately.

Captions for Figures 1, 2, and 3: Please provide the habitat types for each of the figures. Also include the location in the caption for Figure 2.

4. Literature Cited

Falkowski, P.G., and Raven, J.A. (1997). Aquatic photosynthesis. Princeton University Press. Note: Page numbers will differ for the second edition.

---

## Author Response (AR1)

**Reviewer 1**
**This is a well-written, interesting technical note but the data analysis needs further explanations. In this study, the authors use the underwater eddy covariance technique to measure oxygen flux in shallow coastal environments where light reaches the seafloor. From these fluxes, they compute hourly and daily light-use efficiency of the phototrophic benthic community. One of the key findings is that the hourly light-use efficiency may approach the maximum theoretical limit and that it decreases rapidly towards the middle of the day. These are nice results that are also supported by previous work by Berg and colleagues and should be of interest for the readers of Biogeosciences. Light use efficiency is a useful parameter for characterizing and comparing shallow benthic habitats and for assessing environmental change. In a time when coastal water quality is deteriorating globally, a technique allow evaluation of the activity of the phototrophic benthic community is very helpful. I propose expanding the discussion of the calculations of GPP and R and their limitations.**
**Gross primary production (GPP, here total oxygen produced through photosynthesis) was calculated as the sum of the daytime measured net oxygen production and the oxygen consumed through respiration (R) at night. As pointed out by the authors, daytime respiration typically exceeds nighttime respiration, but daytime respiration could not be measured directly in this study. Thus, four different daytime respiration rates were calculated, two static rates and two dynamic rates (linear or sigmoid increases) to determine the respiration behavior that would fit best with the measured data. The accuracy of the determination of R and GPP defines the quality of the light use efficiency estimates that are at the center of this study. In a tidal regime, the eddy covariance instrument may not interrogate the same area of the seafloor during day and night, and thereby produce nighttime R data that are not representative, even after some corrections, of the area producing the daytime flux data. The actual differences in R may be small, however, R then represents a best guess, not a known flux. Another point that could be addressed in more detail are the other controlling factors of benthic photosynthesis besides light intensity, e.g. the spectral composition of the light, roles of grazers, nutrient availability, temperature and current strength.**

Author response: Thank you for taking the time to review our paper. We appreciate the comments and we agree with these two points- as such we are happy to implement these suggestions.
The dataset from Greenland is from a tidal embayment with muddy sediments. The embayment has semidiurnal tides i.e. two high and two low tides every day, so we do interrogate different parts of the seafloor throughout the day. In the mussel reef from the Baltic Sea, the flow direction is less variable since it is determined by large-scale atmospheric patterns. The convention when deriving daily rates as well as P-I relationships using eddy covariance is to assume no significant horizontal flux divergence since the measurements integrate over the small-scale patchiness (Rheuban & Berg 2013). However, we appreciate that this may add variability to our data and in the revised paper we will include an analysis on direction-dependence. We will also expand our discussion on controlling factors of benthic photosynthesis by (1) including an analysis on flow-flux relationships for our datasets, and (2) referencing other studies on other controlling factors.

Cited literature:
Rheuban J.E. and Berg P. 2013. The effects of spatial and temporal variability at the sediment surface on aquatic eddy correlation flux measurements. Limnol. Oceanogr.: Methods 11: 351–359, doi:10.4319/lom.2013.11.351

Implemented changes: We now include a new figure (Fig. 2) illustrating the impact of directionality on the EC fluxes as well as new sections in the Methods (Page 5 L11-17) and Results (3.1 Effects of flow direction; Page 9 L2-7). Overall, we find only a small effect of direction on the fluxes, suggesting that the

eddy covariance measurements adequately integrated over habitat patchiness. As regards to the suggestion to highlight other factors that may influence *GPP*, we now include a sentence in the Conclusion section about this (Page 13 L12-14) and we cite four studies that look at this specifically.

**In figure 1, the data could be interpreted differently, i.e. further increase of the light saturation curves with increasing light. These are four consecutive days of measurements, and the curves of the third and fourth days increase until 300 PAR at least if not farther.**
Author response: The main purpose of this figure is to illustrate that there is no significant flux hysteresis in this dataset. In our revision we will expand our description of day-to-day variations in the light-saturation curves and the P-I fitting parameters, as suggested. We will mention how the parameters change in relation to light availability. In addition to the reviewer suggestions we will also add that day 2 has the lowest Ik and highest alpha, indicating a potential low light acclimation.

Implemented changes: We have expanded our description of the light-saturation curves and their fitting parameters, as requested (Page 9 L18-19; Page 10 L1-2).

**In figure 2, second panel, N1+N2 should be changed to be (N1+N2) average.**
Author response: Thank you for catching this, we will correct it in our revision.

Implemented changes: This has been corrected.

**Although R is about 20% higher in plot 2 of figure 2, GPP is almost identical, and an explanation for this unexpected behavior would be helpful. Similarly, as R increases over time in fig. 2, c and c, one could expect that the curvature of the light-saturation curves would increase but it does the opposite.**
Author response: Figure 2b: Thank you for catching this; we did a mistake in the calculation and offset the daytime fluxes by 3.01 instead of 3.11. We will correct this in the revision and recalculate the P-I relationship.
Fig 2c+d: The curvature does indeed increase compared to panels a+b: the light-saturation parameter $I_k$ decreases, and the alpha increases, by ~20%. This indicates that the curve becomes less linear-like following the correction, which is what we would expect when we correctly account for the marginal hysteresis encountered.

Implemented changes: We have now corrected the mistake in our calculation and recalculated the P-I relationship for this dataset. We also clarify that the curvature does increase following the correction (Page 10 L10-12).

**All four GPP plots in fig. 2 are nearly identical suggesting that magnitude and dynamics of R have little influence on GPP. This is counterintuitive as in many coastal environments R reaches similar magnitude as GPP (as also seen in figure 2) and also follows dynamics that may be similar to those of the GPP (as in fig. 2 d). This needs more detailed explanation.**

Author response: It is true that in this dataset, there is a relatively low impact of light hysteresis on the $O_2$ fluxes. Other eddy covariance studies have documented much larger effects (e.g. Rheuban et al. 2014 Fig. 6). Despite having collected very many datasets in different settings, flux hysteresis is not prevalent in our data, and this is one of the best examples we could find. We will clarify this point in the revised document. Having said that, the exercise in Fig. 2 indicates that hysteresis does have an effect on the P-I relationships. The fitting parameters Ik and alpha hold real-world significance- they represent the photosynthetic performance of the benthic community, so any biases should be accounted for as much as possible.

Implemented changes: We now clarify that the curvature does increase following the correction, and that other studies have found much larger hysteresis effects on the O2 fluxes (Page 10 L12-15).

Cited literature:

Rheuban J.E., Berg P., McGlathery K.J. 2014. Multiple timescale processes drive ecosystem metabolism in eelgrass (Zostera marina) meadows. Marine Ecology Progress Series 507: 1–13. doi: 10.3354/meps10843

**Reviewer 2**
**1. General comments**
**This Technical Note presents a technique that is novel to aquatic ecosystem research and may be highly useful. The approach is simple, but appears to be adequate for this first application. The manuscript is well written and concise, but minor changes can improve it. Specifically, the introduction could be improved by focusing, from the first sentence, on the scientific value that light use efficiency measurements can provide. This can be accomplished largely by re-arranging text, but it will also require more citations. In addition, the Boltzmann equation can be more clearly presented. Finally, two additional figures would also improve the manuscript. The first would be of the sigmoidal fit to PAR data. The second would be of a diurnal time series of light use efficiency. A brief discussion of the data quality, patterns, and implications of that last, proposed, figure would be a valuable addition.**

Author response: Thank you for taking the time to review our paper. We appreciate the comments and there are some very good suggestions in here which we will implement as best we can. Please find our point-by-point response below.

**2. Specific comments**
**2.1 Introduction**
**The introduction focuses on the eddy covariance technique, but LUE is important well beyond eddy covariance. The manuscript would address a larger audience if Section 1.3 can be adapted so that it is a suitable first section. The approach you suggest is exciting. It can be used to investigate the physiological and environmental limitation of photosynthetic production. Those measurements will be of interest outside the eddy covariance community. Former Sections 1.1 and 1.2, that focus on eddy covariance, can be adapted to follow the new Section 1.1. Those sections would show that the tight relationship of eddy covariance measurements to PAR suggests that they will be adequate to resolve LUE in aquatic ecosystems. In the new first section, please also introduce quantum yield measurements. Use caution. In aquatic literature, the term commonly refers to the quantum yield of the photochemistry of photosystem II. Those measurements don't directly compare to LUE. However, there has also been extensive research on the quantum yield of phytoplankton photosynthetic production. Those measurements can be compared directly to your own. Both measurements can reveal environmental and physiological limitations of photosynthesis. Falkowski and Raven (1997) provide a useful summary in Chapter 3.**

Author response: We agree that light-use efficiency is important beyond eddy covariance and we will follow the suggestion to move section 1.3 to 1.1 to make the paper attractive to a broader audience. This is a good suggestion. The original sections 1.1 and 1.2 will then follow. In the new section 1.1 we will introduce the quantum yield of phytoplankton photosynthetic production as a comparative measure, as suggested, and we will compare published values to our own in the Results & Discussion section.

Implemented changes: We have moved section 1.3 to 1.1. We now mention that LUE concepts can be applied to both pelagic and benthic studies and we have introduced the quantum yield of phytoplankton photosynthetic production (Page 2 L 3-7). We now cite a phytoplankton study in the Results and discussion (Page 11 L7-8) that similarly found high LUE under light-limiting conditions.

**2.2 Materials and methods**
**There are four cumulative PAR terms in the equation (left side, A1, A2, and PAR). For clarity, I suggest writing that A1, A2, xo and dx are all fitting parameters. PAR on the left and PAR on the right can be discriminated with different subscripts. Preferably, the subscript would make it clear that one is observed and the other is predicted. I also suggest that x0 and dx would be more clearly represented with as "t" instead of "x," because they are measures of time.**

**To clarify how the sigmoidal fit is used, please include a figure that shows the fit to cumulative PAR. With an additional y-axis, this figure could also show a representative increase in daytime R that is predicted with it. Accordingly, please also include evidence that N2 often exceeds N1.**

Author response: These are good suggestions- we will implement all of them. We will clarify that A1, A2, x0 and dx are all fitting parameters, we will distinguish between observed and predicted PAR using subscripts, and we will replace the 'x' with a 't'. We will also include a figure (a new Fig. 1) showing the fit to cumulative PAR. We will show actual numbers for N1 and N2 for all 24 h sections illustrated.

Implemented changes: We have clarified that A1, A2, x0 and dx are all fitting parameters, and we now distinguish between observed and predicted PAR using subscripts (Page 6 L16). We have replaced 'dx' with 'dt' and we include a new figure (Fig. 1) illustrating the process of using the Boltzmann function to fit cumulative PAR and determine daytime R. We also include values for N1 and N2 flux periods for the datasets showing a minor hysteresis (Fig. 5).

**2.3 Results and discussion**
**Please also add a figure to show a diurnal time series of changes in LUE alongside PAR and GPP. Ik is the irradiance at which the rate of photon absorption matches the maximum turnover rate of photochemistry (Falkowski and Raven, p. 200). Your results can be used to examine if Ik is a point of interest in the diminishment LUE as PAR increases. Assuming the resulting figure is noisy, please also consider techniques that could improve resolution of diurnal variation in LUE.**

Author response: OK, we will add another set of panels for both datasets (Figs. 1+3) showing LUE as a function of time, indicating the corresponding Ik value for each dataset. For illustration purposes we will plot hourly LUE using a log y-axis as shown in Fig. 4.

Implemented changes: We have added another set of panels for both datasets illustrating diurnal hourly LUE (Figs 3, 5). We have not added the Ik value to this plot because the Ik value is shown in the light-saturation curve panel beneath the LUE.

**3. Technical corrections**
**Page 2 Line 20: "offsetting daytime fluxes by the dark rate." "Offsetting" is a vague term. Can you use a more specific one? I suggest "sum" or "summation." You use "offset" again on page 5, line 11.**
Author response: Yes, we will correct these sentences to say that they were computed as a sum of dark and light fluxes.

Implemented changes: We now state that they were computed as a sum of dark and light fluxes (Page 4 L2).

**Page 3 line 7: "magnitude of hysteresis is related to light history." "light history" is also vague term. Could you be more specific?**
Author response: Yes, we will clarify that 'light history' as defined in the paper by Adams et al. (2016) refers specifically to the lag in the ecosystem's response (in terms of $O_2$ production through GPP) to changing light levels.

Implemented changes: We now state that the study by Adams et al. (2016) refers specifically to the lag in the ecosystem's response to changing PAR (Page 4 L10-11).

**Page 5 Defining a daytime R rate. It is not clear that N1 and N2 are measurements of flux, as opposed to periods of time. I suggest clarifying that they are flux. By convention, J is often used for flux.**

Author response: We will clarify that N1 and N2 represent flux measurements corresponding to different time periods of the day (Page 5, L8). Time periods will be denoted as N1 and N2, whereas fluxes corresponding to those time periods will be $J_{N1}$ and $J_{N2}$.

Implemented changes: Here and throughout the paper, we now denote nighttime flux periods N1 and N2 as $J_{N1}$ and $J_{N2}$ (e.g. Page 6 L2).

**Page 5 line 17: "...whereas the fourth approach assumed a sigmoidal increase with time." I'd suggest more specifics here too. Perhaps "...whereas the fourth approach assumed that R increased with cumulative PAR. This was represented with a sigmoidal increase..."**

Author response: OK, we will implement this suggestion.

Implemented changes: We now state specifically that R increased with cumulative PAR, and that this was represented as a sigmoidal increase (Page 6 L11-12).

**Page 7 line 10: Instead of "assuming," I suggest "the assumption that."**

Author response: OK, we will implement this suggestion.

Implemented changes: This has been implemented (Page 8 L6).

**Page 7 line 21 PARhourly *fAPAR should be enclosed by parentheses. Again on page 8, line 2.**

Author response: Thank you, we will implement this suggestion.

Implemented changes: These changes have been implemented.

**Page 8, line 3 Check the cases of section headings. You may want "Results and discussion"**

Author response: OK, we will use lower case 'd' for 'discussion'.

Implemented changes: This was implemented.

**Page 8, line 5 Please describe the habitat types of the dataset from Greenland.**

Author response: OK, we will add a few sentences describing the study site in Greenland, something like this: 'The study site in Greenland is a protected inlet of ~3 km2 with silt-sand sediments and tidally-driven flow velocities typically ranging from 2-10 cm s-1 near the seabed. The instrument was deployed at 3 m water depth.'

Implemented changes: We have added a few sentences describing the study site in Greenland (Page 4 L20-21; Page 5 L1-2).

**Page 9 line 11: In Figure 4 there are a handful of measurements with a very low LUE (0.001 to 0.004). Could you say a few words about these?**

Author response: Yes, in the revision we will include a few sentences about these measurements. In general, we would expect higher LUE under low irradiance. In the 72 h-long dataset from the Baltic Sea, 5 of the 1-h fluxes had similar or lower LUE compared to the rest of the day. Following comments from Reviewer 1, we will include an analysis on the direction-dependence of the fluxes to establish whether they originate from a different part of the reef. We will highlight to what extent this may reflect GPP measurement error or environmental factors such as nutrient limitation at this rocky site.

Implemented changes: We have added a section in the Results and discussion describing these low values (Page 11 L9-14). We do not think this is due to a direction dependence of the fluxes, but rather may reflect differences in how these habitats function with respect to GPP (sedimentary versus rocky).

**Page 9, line 19. Replace "_" with approximately.**

Author response: OK we will replace this.

Implemented changes: This has been replaced.

**Captions for Figures 1, 2, and 3: Please provide the habitat types for each of the figures. Also include the location in the caption for Figure 2.**

Author response: OK we will provide habitat type and location as requested.

Implemented changes: We now provide habitat type and location for each of (new) figures 3, 4 and 5.

**4. Literature Cited**
**Falkowski, P.G., and Raven, J.A. (1997). Aquatic photosynthesis. Princeton University**
**Press. Note: Page numbers will differ for the second edition.**